# Antioxidant Response of Maternal and Fetal Rat Liver to Selenium Nanoparticle Supplementation Compared to Sodium Selenite: Sex Differences between Fetuses

**DOI:** 10.3390/antiox13070756

**Published:** 2024-06-21

**Authors:** Milica Manojlović-Stojanoski, Slavica Borković-Mitić, Nataša Nestorović, Nataša Ristić, Radomir Stefanović, Magdalena Stevanović, Nenad Filipović, Aleksandar Stojsavljević, Slađan Pavlović

**Affiliations:** 1Institute for Biological Research “Siniša Stanković”—National Institute of the Republic of Serbia, University of Belgrade, Bulevar Despota Stefana 142, 11108 Belgrade, Serbia; borkos@ibiss.bg.ac.rs (S.B.-M.); rnata@ibiss.bg.ac.rs (N.N.); negicn@ibiss.bg.ac.rs (N.R.); sladjan@ibiss.bg.ac.rs (S.P.); 2Department of Pathology and Medical Citology, University Clinical Center of Serbia, Pasterova 2, 11000 Belgrade, Serbia; r_stefanovic@hotmail.com; 3Faculty of Medicine, University of Belgrade, dr Koste Todorovića 26, 11000 Belgrde, Serbia; 4Group for Biomedical Engineering and Nanobiotechnology, Institute of Technical Sciences of the Serbian Academy of Sciences and Arts (SASA), Kneza Mihaila 35/IV, 11000 Belgrade, Serbia; magdalena.stevanovic@itn.sanu.ac.rs (M.S.); nenad.filipovic@itn.sanu.ac.rs (N.F.); 5Innovative Centre, Faculty of Chemistry, University of Belgrade, Studentski trg 12-16, 11000 Belgrade, Serbia; aleksandars@chem.bg.ac.rs

**Keywords:** antioxidant response, selenium nanoparticles, sodium selenite, pregnancy, fetuses, liver, histology, oxidative stress parameters, sex differences

## Abstract

To compare the effects of organic selenium nanoparticles (SeNPs, Se^0^) and inorganic sodium selenite (NaSe, Na_2_SeO_3_, Se^4+^) on the antioxidant response in maternal and fetal rat liver, pregnant females were treated with two forms of selenium (Se) at equivalent doses during gestation (0.5 mg SeNPs or 0.5 mg NaSe/kg body weight/day). Structural parameters of the liver of gravid females and their fetuses were examined in a sex-specific manner. The oxidative stress parameters superoxide dismutase (SOD), catalase (CAT), glutathione peroxidase (GSH-Px), glutathione reductase (GR), glutathione S-transferase (GST), total glutathione (GSH) and sulfhydryl groups (SH) were established. In addition, the Se concentration was determined in the blood, liver, urine and feces of the gravid females and in the liver of the fetuses. The structure of the liver of gravid females remained histologically the same after supplementation with both forms of Se, while the oxidative stress in the liver was significantly lower after the use of SeNPs compared to NaSe. Immaturity of fetal antioxidant defenses and sex specificity were demonstrated. This study provides a detailed insight into the differences in the bioavailability of the nano form of Se compared to sodium selenite in the livers of pregnant females and fetuses.

## 1. Introduction

Selenium (Se) is a natural metalloid, an essential micronutrient for humans, which is mainly absorbed through food or food supplements. Selenium deficiency affects between 500 million and 1 billion people worldwide, primarily due to inadequate dietary intake. Selenium levels are significantly lower in many parts of Europe than in the United States, with average Se intake lower in Eastern Europe than in Western Europe [1]. In nature, Se occurs as an analog of sulfur in four oxidation states: Se^6+^ (selenate), Se^4+^ (selenite), Se^2−^ (selenide) and Se^0^ (elemental Se). In food, Se is found in an organic (selenomethionine and selenocysteine) and inorganic form. The biological and toxicological effects of Se, whether in anthropogenic or natural environments, depend on a specific chemical state and their composition [2]. The most commonly used Se formulation is sodium selenite (Na_2_SeO_3_, NaSe, Se^4+^), which has very good absorption. The zero-oxidation state of Se (Se^0^) in selenium nanoparticles (SeNPs) exhibits lower toxicity and better bioavailability compared to other oxidation states [3]. SeNPs showed excellent bioavailability compared to organic Se-methyl-selenocysteine and NaSe, but also equivalent efficacy in activating selenoenzymes [4,5,6]. In adult mice and rats, the use of SeNPs reduced the risk of Se toxicity based on the mean lethal dose, survival rate, absence of histopathological changes in the liver, kidneys and lungs, and biochemical blood parameters [4,7].

Selenium is associated with amino acids, proteins and metabolic intermediates [8]. More than 40 selenoprotein genes have been identified in various tissues in different organisms, including the proposed 25 selenoprotein genes in mammals [9]. In general, all Se-containing proteins are involved in scavenging reactive oxygen species (ROS), including the transport selenoprotein P (SP), which efficiently supplies Se to extrahepatic tissues with ten selenocysteine residues simultaneously [10]. Selenoproteins without known functions are designated with the symbols SEL or SEP [11]. The redox balance in the liver is maintained by the scavenging of ROS by antioxidant defense enzymes such as superoxide dismutase (SOD), catalase (CAT), glutathione peroxidase (GSH-Px), glutathione reductase (GR) and glutathione S-transferase (GST). GSH-Px is a selenoenzyme with different forms: GSH-Px1, GSH-Px2, GSH-Px3, GSH-Px4 and GSH-Px6. In addition, there are many vital selenoenzymes, such as thioredoxin reductases (TXNRD1, TXNRD2 and TXNRD3), iodothyronine deiodinases (DIO1, DIO2 and DIO3), methionine- R-sulfoxide reductase 1 (MSRB1) and selenophosphate synthetase (SEPHS2) [11].

The conversion of Se ingested with food into metabolically active species takes place mainly in the liver. The intracellular cycle of Se to selenide and monoselenophosphate maintains the continuous synthesis of selenoproteins, including selenoenzymes and transport proteins which are secreted into the bloodstream [12]. The expression of SP receptors in the brain, testes, kidneys and muscles ensures the distribution of Se under Se deficiency conditions and creates a tissue hierarchy of Se supply and utilization, with priority given to the brain, reproductive organs and endocrine glands [13]. The excretion of Se via small molecule excretory forms synthesized in the liver makes the liver the central organ for the regulation of Se metabolism, maintaining the required Se concentration range that allows optimal expression of selenoproteins in the organism [10]. Otherwise, reduced or excessive Se intake is associated with adverse health effects [14]. Selenium deficiency is associated with an increased risk of oxidative stress and cancer and impairs reproduction and fertility, while the consequences of Se toxicity (selenosis) can lead to fatigue, neurotoxicity and embryonic malformations [15]. Many studies have focused on investigating the link between oxidative stress as a trigger of genetic and epigenetic dysregulation of oncogenes that can promote neoplastic transformation and tumorigenesis. An important modulator in the response to oxidative stress is nuclear factor erythroid 2-related factor 2 (NRF2), which induces the expression of antioxidant enzymes and thus protects cells from oxidative damage. The NRF2/KEAP1 signaling pathway in cervical and endometrial cancer and chemotherapy resistance can be modulated by various natural and synthetic antioxidant modulators [16]. The same authors [17] suggested a wide range of natural modulators of NRF2, such as diosmetin, wedelolactone, epoximicin, ailantone, Vit K3, caffeic acid and many others. Therefore, treatments with natural or synthetic compounds that can inhibit NRF2 signaling could be an important clinical step to combat or suppress drug resistance in ovarian cancer patients [17]. Selenium exhibits a broad spectrum of contradictory properties in living organisms. The chemopreventive, cancer-promoting and anticancer effects of Se are three different aspects of the complicated relationship of Se to cancer [18], which likely depends on its chemical form and bioavailability. In addition, pre-eclampsia, which can occur during pregnancy and has a direct effect on the outcome of the pregnancy, is also characterized by increased oxidative stress and inflammation, in which NRF2 also plays a key role [19]. The study by Vanderlelie and Perkins [20] has shown that Se supplementation can help reduce oxidative stress in women at risk of pre-eclampsia. Adequate maternal Se intake during pregnancy is crucial for normal prenatal development and lays the foundation for the development of the mental and physical health of the offspring [21]. Substances with antioxidant properties have been shown to play a key role in the treatment of diseases associated with oxidative stress. Therefore, the development of nano formulations based on biopolymers with antioxidant properties represents the future in this field of research. An example of this is the development of nano-drug systems based on biopolymers with antioxidant properties, which offer significant potential to improve drug delivery efficiency and therapeutic outcomes. Despite various obstacles, such as biocompatibility, stability and scalability, research in this area offers promising results [22].

The present study aimed to compare the mode of action of Se in the liver during pregnancy with two different dietary forms of Se: bovine serum albumin (BSA)-stabilized Se nanoparticles (SeNPs, Se^0^) and inorganic sodium selenite (NaSe, Na_2_SeO_3_, Se^4+^). The two forms of Se were administered in equivalent doses throughout pregnancy (0.5 mg SeNPs or 0.5 mg NaSe kg/body weight/day). The structural and functional parameters of the liver of gravid females and their fetuses (females vs. males) were investigated. In addition, we determined the antioxidant defense enzymes SOD (EC 1.15.1.1), CAT (EC 1.11.1.6), GSH-Px (EC 1.11.1.9), GR (EC 1.6.4.2) and GST (EC 2.5.1.18), as well as the low molecular weight antioxidants total glutathione (GSH) and sulfhydryl groups (SH). Selenium concentrations were determined in the blood, urine and feces of adult females and in the liver of pregnant females and fetuses.

## 2. Materials and Methods

### 2.1. Animals and Experimental Groups

The experiments were performed in accordance with the OECD Guidelines for Testing Chemicals for a Prenatal Developmental Toxicity Study (Test No. 414, adopted 14 June 2018) [23] on adult female rats of the Wistar strain (Rattus norvegicus) weighing approximately 220–250 g, bred at the Institute for Biological Research “Siniša Stanković”—National Institute of the Republic of Serbia, University of Belgrade, Belgrade, Serbia. The animals were kept under controlled conditions (12 h light–dark cycle at 22 °C) and had ad libitum access to food (standard rat chow) and tap water. The full chemical composition of the rat diet is shown in Table 1.

According to ethical standards, all animal experiments complied with the Council Directive of the European Communities (86/609/EEC) and were approved by the Ethics Committee for the Use of Laboratory Animals of the Institute for Biological Research “Siniša Stanković”—National Institute of the Republic of Serbia, University of Belgrade, Serbia (No. 2-12/13). The experiments were conducted in accordance with the guidelines of ARRIVE (Animal Research: Reporting in Vivo Experiments) and the Guidelines on the Principles of Regulatory Approval of 3R (Replacement, Reduction and Refinement) Test Approaches.

Two nulliparous females in estrus were mated with a fertile male in separate cages. The day on which sperm was detected in the vaginal smear was designated as day 0 of pregnancy. Each experimental group consisted of six sperm-positive females, which were randomly assigned into: 1. intact control group (IC); 2. control group (C) administered BSA and vitamin C by gastric gavage; 3. group with SeNPs administered at a dose of 0.5 mg SeNPs/kg/body weight; and 4. group with NaSe administered at a dose of 0.5 mg Na_2_SeO_3_/kg body weight/day. The administered volume was 0.2 mL in all groups. The administered dose corresponds to the amount of Se consumed in the diet for which a spectrum of positive effects without undesirable side effects has been demonstrated [7,24,25]. Treatment of the dams by gastric gavage began on day 1 of gestation and lasted until day 20 of gestation. On day 21 of gestation, the pregnant dams were sacrificed by decapitation with the Harvard guillotine and the fetuses were removed from the uterus, referred to as 21-day-old fetuses. The average number of fetuses per dam was calculated to be 12 ± 2, with an even distribution of the sexes, the average number of male and female fetuses being 6 ± 1. Pregnant dams and fetuses were dissected within 3 min, and the isolated livers were placed in ice-cold 155 mmol/L NaCl and washed with the same solution. After isolation, the liver tissue was frozen in liquid nitrogen (−196 °C) and stored at −80 °C until oxidative stress parameters analysis, or fixed in 4% paraformaldehyde for further histologic analysis. The livers, blood, urine and feces of gravid females were stored at −80 °C for determination of Se concentration. The sex of the fetuses was determined during dissection based on the presence of ovaries or testes in the abdominal cavity. Four experimental groups of male and four experimental groups of female fetuses were formed (IC, C, SeNPs and NaSe), each containing 10 fetuses, taking care to ensure that the samples (for the determination of oxidative stress parameters, Se concentration and histological analysis) within each group were formed from fetuses originating from different mothers.

Special care was taken to minimize animal suffering during daily handling and sacrifice.

### 2.2. Synthesis and Physicochemical Characterization of SeNPs

The SeNPs were synthesized by chemical reduction of sodium selenite as described in our previous paper [26]. In summary, the solution of this Se compound was slowly added to a vessel containing a mixture of dissolved Vit C and BSA. After the colorless reaction mixture turned brick red, indicating the formation of SeNPs, the resulting colloidal solution was filtered through a 0.24 μm syringe filter (Millipore, Burlington, MA, USA) and stored in a refrigerator. Due to the photosensitivity of Se, the reaction of SeNPs formation was carried out in a coated vessel.

The size and morphology of the synthesized SeNPs were examined using a transmission electron microscope (TEM) (JEM 2100, JEOL, Tokyo, Japan). The colloidal solution of SeNPs was placed dropwise on the carbon film supported by a 300-mesh copper grid and allowed to dry at ambient conditions before being placed in the microscope holder. After visualizing the particles, selected area electron diffraction (SAED) was performed to investigate the crystalline nature of the sample.

The size distribution of SeNPs within the larger volume of the sample was determined with a particle size analyzer [22,27] using the Hydro 2000 μP wet sample dispersing unit (Mastersizer 2000, Malvern Instruments Ltd., Malvern, UK). Qualitative characterization of the SeNPs was performed by Raman spectroscopy. The Raman spectra were recorded using a Thermo Scientific DXR Raman microscope equipped with a HeNe laser (Waltham, MA, USA). A small amount of SeNPs in the form of a colloidal solution was placed on the sample holder and the measurement was performed under the following conditions: excitation at 633 nm with a constant laser power of 8 mW, a grating of 600 lines/mm and a spectrograph aperture of 50 μm. The exposure time of the sample was 60 s and the number of scans was 10. Fluorescence correction was performed by fifth-order polynomial fitting (OMNIC v8.0 software for DXR Dispersive Raman).

Quantitative analysis of Se in the SeNPs was performed by inductively coupled plasma optical emission spectrometry (ICP-OES) (Thermo Scientific iCap 6500 Duo, Waltham, MA, USA). The measurement was performed with the prepared colloidal solution of SeNPs, which was first filtered through a 0.44 μm syringe, Millipore, Merck, Burlington, MA, USA) and then diluted with 2.5% nitric acid. The intensity of the emission was measured at λ = 196.26 nm. Selenium from the standard solution MES-21-1 (AccuStandrad, New Haven, CT, USA) was used as a calibration standard.

### 2.3. Histological Analysis

For histologic analysis and light microscopy, a portion of the left median lobe of the liver was removed from the mothers and their fetuses and fixed for 24 h in a 4% paraformaldehyde solution in phosphate-buffered saline with a pH of 7.4, dehydrated with increasing concentrations of ethanol and xylene, and embedded in paraffin. Five-micrometer-thick tissue sections were deparaffinized in xylol and dehydrated with decreasing concentrations of alcohol. Liver sections from gravid females were stained with Masson’s trichrome, while fetal liver sections were stained with hematoxylin and eosin (H&E). Image acquisition and histologic analysis were performed using a microscope (Olympus, BX-51, Olympus, Tokyo, Japan) equipped with a CCD video camera (PixeLink, Pix eLINK, Gloucester, ON, Canada) and image acquisition and analysis software (Visiopharm Integrator System (VIS), ver. 2020.01.3.7887; Visiopharm, Hørsholm, Denmark).

### 2.4. Preparation of Liver Tissues and Determination of Oxidative Stress Parameters 

Liver tissue samples collected from mothers and fetuses were washed in 0.9% NaCl, immediately frozen in liquid nitrogen (−196 °C) and stored at −80 °C for no longer than one month until analysis. Tissues were minced and homogenized on ice in 10 volumes of 25 mmol/L sucrose with 10 mmol/LTris-HCl, pH 7.5, supplemented with 1 × phosphatase inhibitor mix I and 1 × protease inhibitor mix G at 4 °C [28] using an IKA-Werk Ultra-Turrax homogenizer (Janke and Kunkel, Staufen, Germany) [29]. The homogenates were sonicated on ice at 10 kHz for 30 s (Bandeline, Sonopuls, Berlin, Germany) to release enzymes [30], followed by centrifugation in a Beckman ultracentrifuge at 100,000× *g* for 90 min at 4 °C [31,32].

The activity of superoxide dismutase (SOD, EC 1.15.1.1) was determined using the epinephrine method [33]. One unit of SOD activity was defined as the amount of protein causing 50% inhibition of the autoxidation of epinephrine at 26 °C and was expressed as U/g wet mass. Catalase (CAT) activity (EC 1.11.1.6) was determined by the rate of hydrogen peroxide (H_2_O_2_) degradation [34] and expressed as µmol H_2_O_2_/min/g wet mass. The activity of glutathione peroxidase (GSH-Px, EC 1.11.1.9) was determined by following the oxidation of nicotinamide adenine dinucleotide phosphate (NADPH), which serves as a substrate, with t-butyl hydroperoxide [35] and expressed as nmol NADPH/min/g wet mass. Glutathione reductase (GR, EC 1.8.1.7) activity was measured according to the method of [36], which is based on the ability of GR to catalyze the reduction of oxidized glutathione (GSSG) to reduced glutathione (GSH) using NADPH as a substrate in a phosphate buffer (pH 7.4). GR activity was expressed as nmol NADPH/min/g wet mass. The activity of glutathione S-transferase (GST, EC 2.5.1.18) toward 1-chloro-2,4-dinitrobenzene (CDNB) was determined according to the method of [37] and expressed as nmol GSH/min/g wet mass. The method is based on the reaction of CDNB with the SH group of GSH catalyzed by the GST present in the samples. Antioxidant enzyme activities were determined simultaneously in triplicate for each sample using a Shimadzu UV-1900i spectrophotometer with a temperature-controlled cuvette holder at 37 °C (Shimadzu Corporation, Nishinokyo-Kuwabaracho, Nakagyo-ku, Kyoto, Japan), and the mean values of three measurements were used for further calculation. All chemicals were purchased from Sigma-Aldrich (St. Louis, MO, USA).

Total GSH concentration was determined as described by [38] and expressed in nmol/g tissue. Concentrations of SH groups were determined with DTNB according to the method described by [39] and expressed in µmol/g wet mass.

### 2.5. Determination of Se Concentration in the Liver, Blood, Urine and Feces of Pregnant Females and Fetuses

Approximately 0.4 g wet mass of the sample was weighed with an analytical balance and transferred to a microwave vessel. Subsequently, 1 mL of hydrogen peroxide (30% H_2_O_2_, high purity, Merck Millipore, MA, USA) and 2.0 mL of nitric acid (67–69% HNO_3_, high purity, Merck Millipore, MA, USA) were added. Samples were subjected to microwave digestion in a closed vessel using the ETHOS 1 Advanced Microwave Digestion System (Milestone, Milan, Italy). The digestion protocol included ramping to 200 °C for 15 min, holding the temperature at 200 °C for 20 min and cooling to room temperature. After digestion, the clear solutions were carefully transferred to 25 mL polyethylene tubes and diluted to a final volume of 25 mL. Dilution was performed with Milli-Q water (18.2 MΩ × cm) from a Milli-Q Plus system (Merck, Darmstadt, Germany).

The urine samples were diluted 5-fold with 0.05% nitric acid and analyzed directly after filtration through HPLC filters (0.45 µm). Selenium was quantified using inductively coupled plasma mass spectrometry (ICP-MS, iCAP Q, Thermo Scientific, Feltham Middlesex, UK). Se was quantified by external calibration using an ICP multi-element standard solution (Merck Millipore, Darmstadt, Germany) covering concentrations from 1 to 100 ng/mL. An internal online standardization (10 ng/mL Ge solution) was used. The validation of the ICP-MS analysis was tested with a certified reference material from Seronorm™ blood (SERO210105-Level-2, Sero AS, Billingstad, Norway), with element recoveries ranging from 97.5 to 103.7%.

### 2.6. Statistical Analysis

Data analysis including mean ± standard deviation (SD) was performed using STATISTICA software (v. 12.5, Paolo Alto, CA, USA). The distribution of the data was checked with the Shapiro–Wilk test (n < 50) [40]. Identification of outliers from nonlinear regression was performed with GraphPad Prism software (v. 5.0., Boston, MA, USA) using the ROUT method based on the false discovery rate (FDR) [41] with Q (maximum desired FDR) = 10%. The difference between pregnant females and fetuses, between treated groups within adult females and fetuses, and between male and female fetuses, was tested using a nonparametric one-way ANOVA and a factorial ANOVA. As interactive effects were observed, Tukey’s HSD (honestly significant difference) post-hoc test for unequal N was used to find significant differences between groups [42]. The differences in Se concentration between the studied groups were tested with the nonparametric Kruskal–Wallis ANOVA and the post-hoc Mann–Whitney U test. The data set of measured oxidative stress parameters was subjected to principal component analysis (PCA) to reduce the number of descriptors responsible for the highest percentage of total variance in the experimental data. Eigenvalues > 1 were used for the extracted factors, while variables with factor loadings > 0.5 were used to interpret the results. For all tests used, *p* < 0.05 was considered statistically significant.

## 3. Results

### 3.1. Results of the Characterization of SeNPs

The mild conditions during the synthesis of SeNPs by chemical reduction often lead to a spherical morphology. On the other hand, parameters such as the concentration ratio between precursors and stabilizers and the choice of the stabilizer itself have a stronger influence on the size and stability of the obtained nanoparticles. Since size, shape and crystallinity can strongly influence the behavior of the nanoparticles in the biological environment, it was essentially necessary to verify these properties of the SeNPs before proceeding with the biological activity test. Based on the TEM image (Figure 1A), the SeNPs are spherical, uniform, have a diameter of less than 100 nm and are amorphous, as shown by the pattern of electrons diffracted by the particles. Measurement of the particle size distribution (Figure 1B) confirms the electron microscopy result, with a narrow peak indicating a uniform size. Although Raman spectroscopy is a less common vibrational spectroscopy technique, it is very useful for the qualitative analysis of inorganic materials, which usually exhibit weak bands in the lower wavenumber range. In the case of Se, characteristic peaks around 230 or 250 cm^−1^ can be recorded, depending on the arrangement of the atoms. The former originate from stretching vibrations of the atoms within the trigonal crystal structure, while the latter is due to the A1 intrachain stretching mode of the amorphous Se atoms [43]. The Raman spectrum of the SeNPs used in this study (Figure 1C) shows that the Se atoms in the nanoparticles have an amorphous arrangement. The more intense peak in the spectrum originates from the sample holder (CaF_2_). The concentration of Se in the colloidal solution of SeNPs was 640 μg/mL according to ICP-OES.

All results indicate that the SeNPs had comparable properties to those used in previous studies.

### 3.2. Histological Analysis

Histologic analysis of the livers of pregnant females showed that neither treatment with SeNPs nor NaSe caused histopathologic changes. As in control females, normal tissue architecture was observed, with the hepatic sheets radiating from the central vein and the liver parenchyma being homogeneous and fine-grained (Figure 2). The livers of 21-day-old rat fetuses about to be born are characterized by loosely arranged, disorganized strands of hepatocytes that are several cells thick and separated by primitive sinusoids (Figure 3). The hepatocytes contain abundant intrahepatocellular glycogen (free space). Extramedullary hematopoiesis is pronounced and consists of islands of erythropoiesis in the sinusoids, scattered megakaryocytes and myelopoiesis around the portal vein. A mature bile duct can be observed near some of the portal veins (Figure 3c). No histologic changes were observed after maternal treatment with SeNPs or NaSe.

### 3.3. Oxidative Stress Parameters

#### 3.3.1. Parameters of Oxidative Stress in the Liver of Pregnant Females

The activity of SOD was significantly reduced in the liver of the NaSe group compared to groups IC and C (*p* < 0.05). The activity of CAT was also significantly decreased in the livers of the NaSe group compared to the other groups, i.e., IC, C and SeNPs (*p* < 0.05). In contrast, the activity of GSH-Px was significantly increased in the livers of the NaSe group compared to the IC, C and SeNPs groups (*p* < 0.05). The activity of GR was significantly increased in the SeNPs and NaSe groups of adult females compared to the IC and C groups (*p* < 0.05). Similarly, the activity of GST was significantly increased in the livers of both the SeNPs and NaSe groups, but only compared to the C group of adult animals (*p* < 0.05). SH levels were significantly decreased in the liver tissue of the NaSe group of rats only compared to the SeNPs group (*p* < 0.05). In addition, there were no changes in GSH levels between the groups of adult animals studied (Table 2).

#### 3.3.2. Parameters of Oxidative Stress in the Liver of Fetuses of Both Sexes

Looking at the livers of fetuses of both sexes, there were no differences between the activities of SOD, GSH-Px and GST between the animal groups studied (Table 2). However, the activities of CAT and GR were significantly increased in the livers of fetuses from the NaSe group compared to the IC, C and SeNPs groups (*p* < 0.05). The concentration of SH groups was significantly increased in the livers of control fetuses compared to intact animals (*p* < 0.05). The concentration of SH groups in the livers of the C group was higher than in the IC group, but it was significantly decreased in the SeNPs fetuses compared to the C group (*p* < 0.05). In the livers of the NaSe group, the concentration of SH groups was significantly decreased compared to the IC, C and SeNPs groups (*p* < 0.05). The GSH concentration was significantly reduced in the fetal livers of all examined animals compared to the intact animals (*p* < 0.05).

#### 3.3.3. Differences in Liver Oxidative Stress Parameters between Pregnant Females and Fetuses of Both Sexes

The activities of all the investigated enzymes and the total GSH concentration in the liver of the fetuses were significantly lower (*p* < 0.05) than in the adult females when comparing the compatible groups (IC vs. C; C vs. C; SeNPs vs. SeNPs; and NaSe vs. NaSe). Interestingly, only the concentration of SH groups was significantly higher (*p* < 0.05) in the liver of fetuses compared to adult individuals between the respective animal groups (Table 2).

#### 3.3.4. Differences in Oxidative Stress Parameters between Female and Male Fetuses

The results of liver oxidative stress parameters between the corresponding groups of male and female fetuses (Table 3) showed a significant decrease in CAT, GSH-Px, GR, GST and GSH in the IC group of female fetuses compared to male fetuses (*p* < 0.05). In the control group, SOD was decreased in the female fetuses, while CAT and GST were increased compared to the male fetuses (*p* < 0.05). In the liver of the female fetuses with SeNPs, the concentrations of SH and GSH were increased, while in the NaSe group, CAT and GSH-Px were decreased compared to the male fetuses (*p* < 0.05).

In the liver of the male fetuses, CAT activity was significantly lower in the control and SeNPs groups than in the intact control (*p* < 0.05), while it was significantly higher in the NaSe group than in the IC, C and SeNPs groups (*p* < 0.05). GSH-Px activity in the liver of the NaSe group was significantly lower than in the IC group and significantly higher than in the C and SeNPs groups (*p* < 0.05). GST activity was significantly reduced in the C, SeNPs and NaSe groups compared to the IC group (*p* < 0.05). The concentration of the SH groups was significantly lower compared to the IC and C groups and significantly higher compared to the SeNPs group (*p* < 0.05). The GSH concentration in the livers of the male fetuses was significantly lower in the C, SeNPs and NaSe groups than in the IC group (*p* < 0.05).

In the livers of the female fetuses, the activities of SOD were significantly lower in the C group than in the IC, SeNPs and NaSe groups (*p* < 0.05). CAT activity was significantly increased in the fetal livers of the C group compared to the IC animals, but it simultaneously decreased compared to the SeNPs and NaSe animals (*p* < 0.05). At the same time, GSH-Px activity was significantly increased in the C and NaSe groups compared to the IC group (*p* < 0.05), as well as in the NaSe animals compared to the SeNPs group. GR activity was increased in the fetal livers of the NaSe group compared to the IC and C groups (*p* < 0.05). The concentration of SH groups was decreased in the SeNPs group compared to the controls and also in the NaSe group compared to the IC and C groups (*p* < 0.05).

### 3.4. Concentration of Se in Biological Samples

#### 3.4.1. Se Concentration in Blood, Urine and Feces of Pregnant Females

As expected, the Se concentration in the blood and urine of the SeNPs group was significantly increased compared to the IC and C groups. In the NaSe group, Se concentrations in the blood and urine were significantly increased compared to the IC group and decreased compared to the SeNPs group (*p* < 0.05). Selenium concentration in the urine was also increased in the controls compared to the intact controls (*p* < 0.05). There were no changes in fecal Se concentrations between the groups of adult female animals studied (Table 4).

#### 3.4.2. Se Concentration in the Liver of Pregnant Females and Fetuses of Both Sexes

As expected, Se concentration was significantly increased in the liver of the SeNPs group compared to IC and C (Table 5). In the NaSe group, Se concentration was significantly increased in the IC group and decreased compared to the SeNPs group (*p* < 0.05). Se concentrations in the liver were also increased in the controls compared to the intact controls (*p* < 0.05).

In the fetuses, Se concentrations were increased in the SeNPs and NaSe groups compared to IC and C, but also in the C group compared to the IC group (*p* < 0.05). In the fetal livers, there were no changes in Se concentration between the SeNPs and NaSe groups. Se concentration in the liver was significantly lower in fetuses of both sexes compared to adults in all groups studied (*p* < 0.05).

#### 3.4.3. Se Concentration in the Liver of Female and Male Fetuses

In the livers of male and female fetuses treated with Se supplements (SeNPs or NaSe), Se concentration was significantly higher than in the IC and C groups (*p* < 0.05), and also in the C group compared to the IC group (Table 6). In contrast to females, Se concentration was higher in males in the SeNPs group than in the NaSe group. Interestingly, the Se concentration was higher in females than in males in the C and NaSe groups (*p* < 0.05).

### 3.5. Principal Component Analysis

#### 3.5.1. Principal Component Analysis in Pregnant Females and Fetuses of Both Sexes

After checking for sample and model adequacy for all groups, preliminary correlation and eigenvalue analysis, three principal components were extracted for each group to define the variable space. For the differences between adult pregnant females and fetuses of both sexes, two of them (Table 7, Figure 4A) accounted for 81.34% of the explained variance, and data reduction was toward the dominance of factor 1 (67.02%), in which the most contributed parameters were SOD, GSH and GR. Factor 2 (14.32%) included SH, GST and GSH-Px as dominant (Table 7, Figure 4A).

#### 3.5.2. Principal Component Analysis in Female and Male Fetuses

After checking for sample and model adequacy for all groups, preliminary correlation and eigenvalue analysis, three principal components were extracted for each group to define the variable space (Table 7, Figure 4B). For the differences between female and male fetuses, two of them accounted for 68.98% of the explained variance, and data reduction went toward the dominance of factor 1 (42.99%), with GSH-Px, GSH and GST as variables that contributed most to the between-group changes. Factor 2 (25.99%) of total variance included SH, CAT and GR as dominant parameters (Table 7, Figure 4B).

#### 3.5.3. Projection of the Cases on the Factor Plane Based on All Examined Parameters between Pregnant Females and Both Female and Male Fetuses

The separation of the investigated groups based on the investigated parameters in adult females and fetuses of both sexes clearly shows the separation of the IC and C groups according to factor 1 (explaining 86.70%) from the other groups, with the exception of the SeNPs group of adult pregnant females, which are also below the axis. According to factor 2 (7.93%), there is a clear separation of the fetal experimental groups on the left side of the axis compared to the adults on the right side of the axis. This clearly shows that the oxidative stress parameters as a whole system are clearly separated between adults and fetuses, but organized in the same way in adults and fetuses, and they respond similarly after treatment (Figure 5A). Treatment with SeNPs brings the system back to control levels in adults compared to fetuses.

#### 3.5.4. Projection of the Cases on the Factor Plane Based on the Examined Parameters between Female and Male Fetuses

To compare how different forms of Se (SeNPs and NaSe) affect the fetal antioxidant defense system, we performed PC analysis and the results show that factor 1 and factor 2 explain 82.22% of the total variance. According to factor 1 (53.97%), female fetuses treated with NaSe are significantly separated from the other groups, while according to factor 2 (28.25%), male fetuses treated with SeNPs are significantly separated, representing a similar response to treatment as in adult females (Figure 5B).

To show the complete organization of the antioxidant system for all treatments and groups together, the results of the PC analysis are shown in the component plot in rotated space separately for adult females, female fetuses and male fetuses (Figure 5B).

## 4. Discussion

Supplementation with different forms of Se increases the Se status in the blood and liver of pregnant females and in the liver of their fetuses. Biomarkers of oxidative stress, including the activities of antioxidant defense enzymes and non-enzymatic antioxidants, were affected by the increased Se concentration. There were marked differences in the response of the pregnant females’ livers in balancing redox reactions to the two administered forms of Se. In parallel with reaching a narrow range of Se concentrations characteristic of safe uptake, the excretion of excess Se in the mother’s urine increased significantly, especially in the SeNPs group. The immaturity of the antioxidant defense system of intact 21-day-old fetuses compared to their mothers was demonstrated, as well as inherent sex differences, with female fetuses exhibiting significantly lower levels of enzymatic and non-enzymatic biomarkers. After Se administration to the mothers, both groups of Se-exposed fetuses were found to have increased Se concentrations in the liver, which affected the antioxidants in their livers, depending on the Se form and the sex of the fetuses.

Adequate maternal Se status during pregnancy is essential for healthy development in utero and for the health and antioxidant protection of the offspring. Under conditions of Se deficiency, the supply to the fetus is significantly reduced and can affect the course of development [44]. These include miscarriages, the risk of premature birth, small for gestational age newborns, a lower psychomotor score in 6-month-old infants and an increased risk of infection in the first 6 weeks of life [45,46,47,48]. New Se formulations could be an efficient way to improve inadequate Se levels during pregnancy, with maximum benefit and minimal adverse effects, both for the mother and the fetus [49], although dosing remains a critical issue in Se use. Administration of both forms of Se, SeNPs and NaSe, at equivalent doses of 0.5 mg/kg body weight, labeled as exceeding dietary requirements, did not cause changes at the histological level in the liver of pregnant females. Our results showing that the fine, healthy structure of the liver lobes was preserved are consistent with the results of numerous studies conducted in adult male rats showing beneficial effects of supplementation with SeNPs in a similar dose range (0.2–0.8 mg/kg body weight) on antioxidant capacity, reproductive function and immunity [5,6,24,25]. Damage to the liver parenchyma, impairment of the intestinal epithelium and chronic toxicity were demonstrated at doses of 1.5 mg/kg, 2 mg/kg, or 4 mg/kg SeNPs, depending on the experimental design and rat strain [5,6,24,25]. In domestic animals (sheep and goats), Se supplementation in the form of nanoparticles in the basal diet has great potential, with doses ranging from 0.3 mg/kg body weight to 3.0 mg/kg body weight supporting better overall animal performance [50,51,52]. For humans, an adequate Se intake level of 70 µg/day, the lowest observed adverse effect of 330 μg/day and an upper intake level of 255 μg/day for adult men and women have been established [53]. Acute and chronic toxicity of dietary Se occurred above 1000 μg/day [54]. The individual requirement for Se varies greatly and it is difficult to determine the toxic level of intake of this trace element, which depends on the combination with other dietary components, especially cations, but also on the genetic variants in selenoprotein genes, i.e., in the organism itself [12].

During pregnancy, the increased proliferative and synthetic activity of the maternal liver is aimed at supporting the development and growth of the embryo/fetus [55,56]. Thus, pregnancy itself can be a cause of oxidative stress, which implies the need for adequate nutritional supplementation with Se [52]. The enterohepatic circulation transports Se-rich blood from the gastrointestinal tract via the portal vein to the liver, where further functionalization of Se takes place. Our results showed that SeNPs supplementation during pregnancy led to a 12-fold increase in circulating Se concentration, while the increase after NaSe was 2-fold. Intestinal absorption of NaSe is efficient, but its clearance appears to be high, resulting in lower retention than organic forms of Se [12]. Spherical, highly water-soluble SeNPs, in which the Se is in the zero redox state, have a significantly higher absorption rate than the inorganic selenite, as shown by our results. This structure of the nanoparticles had a higher surface-area-to-volume ratio, which provides more surface area for the particles to adhere to the mucosa covering the intestinal lumen, increasing the residence time in the intestine, potential interaction with the membrane transport proteins of enterocytes and the possibility of passage into the bloodstream [57]. Similarly, results in Khalkhali goats evaluating the total Se content of whole blood and serum showed that SeNPs were absorbed more efficiently than NaSe [58].

Although the mechanisms of Se accumulation in the liver are not fully understood, our results show that vitamin C as a potent antioxidant, to which the control group was exposed, positively affects the Se status of the liver, and a similar effect was shown at high vitamin C concentrations [59]. Interestingly, treatment with NaSe during pregnancy achieved the same level of Se storage in the liver. At the same time, SeNPs exceeded NaSe-induced accumulation by almost 2-fold, suggesting a role of specific structural features of SeNPs and the positive influence of vitamin C as an SeNPs solvent.

Increased Se concentration in the supplemented dams affected the activities of the liver antioxidant defense system: SeNPs increased the activity of GR and GST, while NaSe affected a broader range of enzymes, i.e., it increased the activity of GSH-Px, GR and GST and decreased the activity of SOD and CAT. These results are in line with the findings of other authors [60] who reported that Se supplementation increased the activity of GSH-Px and the concentration of GSH. Selenium has a similar chemical property to sulfur in cells. This is of immense importance for low molecular weight S-containing antioxidants such as GSH and SH. However, Se has some advantages over sulfur. The main differences lie in the way the element is incorporated into active sites of enzymes. Sulfur as a thiol is incorporated into proteins via cysteine (Cys-SH), whereas Se as selenol is incorporated into proteins as selenocysteine (Cys-SeH). This difference leads to a much greater nucleophilicity of Se-cysteine, which makes it much more reactive toward hydroperoxides than Cys-SH [60].

Thus, the bioavailability of NaSe was increased with respect to the induction of Se-enzymes in the liver compared to SeNPs and its effects. The increased activity of these GSH-associated enzymes and the decreased SH levels could be associated with increased Se-dependent GSH-Px activity and an attempt to restore homeostasis by increasing the levels of reduced components. Indeed, the induction of GR and GST activity is an indicator of GSH recycling and conjugation to various electrophilic compounds to make them more water-soluble and easier to excrete from the body to attenuate oxidative stress and thus contribute to cellular detoxification and maintenance of cellular redox balance [61,62]. Overall, this suggests that NaSe triggers a stronger induction of GR and GST than SeNPs. These results are consistent with the short-term toxicity study of SeNPs and NaSe exposure in mice, in which selenite caused stronger oxidative stress than the SeNPs effect [62]. The significant decrease in SOD and CAT activities in the NaSe group, but not in the SeNPs group, and the GST activity induced by both forms of Se are consistent with our findings on the effects of excessive Se dosing during pregnancy [26]. Gan et al. [63] found that GSH-Px expressions and activities increase with increasing Se exposure in the nutrient range, while overexposure could lead to a decrease in GSH-Px expressions and activities [63]. Microarray studies on the whole liver transcriptome of rats showed that there were no significant changes in gene expression of selenoproteins, including GSH-Px associated with excessive and toxic Se concentrations [64].

The results presented in our study also show that SeNPs offer advantages in terms of toxic effects on the liver due to the lower pro-oxidant effect compared to NaSe. While selenite can generate ROS through redox reactions, for example, the oxidation of thiols can directly lead to the formation of superoxide, and Se in the form of nanoparticles may have lower reactivity and ROS-generating potential due to its particular physicochemical properties [65]. The structure of the SeNPs used, in which the interaction of nano-red elemental Se^0^ with BSA acts as a dispersant, prevents the aggregation of elemental Se to inert, black or gray Se^0^ above a size of 300 nm and ensures its nanoscale size, which prevents it from becoming biologically inert. In addition, the controlled release of Se atoms (Se^0^) from the nanoparticles contributes to their lower pro-oxidant activity, while the surface of Se nanoparticles can scavenge ROS, further reducing their potential to trigger oxidative stress [4]. The toxic effect of Se is inversely proportional to the particle size, so the cytotoxicity of SeNPs (50–100 nm) is lower compared to NaSe (0.1 nm) [66]. In agreement with these results, PCA separated the IC, C and SeNPs groups of pregnant females from the NaSe group and confirmed that the liver response to SeNPs was consistent with the responses of the controls (IC and C). Consequently, the bioavailability, oxidative stress and toxicity of SeNPs were lower than those of NaSe.

In addition, the present results confirm that urinary excretion of excess Se is the main mechanism to avoid toxicity and ensure the amount required for fetal development, as described in the literature [10]. Urinary Se excretion increased 6-fold in the SeNPs group and 1.5-fold in the NaSe group, while fecal excretion of Se was at the control level in these groups. The interaction of the kidneys and liver plays an important role in Se metabolism, as the excretion of small, water-soluble Se excretion molecules synthesized in the liver facilitates the removal of excess Se by the kidneys to prevent oxidative stress and tissue damage and maintain the overall physiological balance in the body [14].

This study provides unique results examining the livers of gravid females and fetuses structurally and functionally from the perspective of the antioxidant defense system and provides valuable insight into maternal–fetal communication, including placental function. In 21-day-old fetuses (the designation is 22nd day), the liver structure has a recognizable lobular arrangement as shown and already performs numerous metabolic functions, including glycogen metabolism, regulation of glucose levels, protein synthesis and detoxification, all of which are essential for supporting fetal growth [67]. Histologic analysis has also shown that hematopoiesis of the liver in 21-day-old fetuses parallels a gradual transition to hematopoiesis of the bone marrow, which is one of the essential aspects of fetal development in adapting to postnatal life [68]. Under these conditions, the antioxidant defense system of the fetal liver is critical in protecting against oxidative stress, maintaining cellular integrity and preventing developmental abnormalities, thereby programming the long-term health outcomes of the offspring.

Comparing mother and fetus, PCA shows that the antioxidant defense system in adults and fetuses is organized according to the same principle and reacts in the same way. On the other hand, a lower level of activity and concentration of the components of the antioxidant defense system was an indicator of the immaturity of antioxidant protection in the fetus, making it more susceptible to oxidative stress, but appropriate for the in utero environment under physiological conditions.

Interestingly, the results show inherent sex differences during the fetal period and demonstrate the lower activity of oxidative stress parameters in the liver in intact females, including CAT, GSH-Px, GR, GST and GSH, than in intact males, given the widely accepted concept that males tend to be more sensitive to various insults during the perinatal period [69]. The “Quiet Embryo Hypothesis”, which states that viable preimplantation embryos have a lower metabolite turnover, indicating a lower need for antioxidant defense systems than their less viable counterparts [70], could be the explanation for the results presented. Indeed, differences in metabolic rate and antioxidant defense parameters during early prenatal development could be the cause of the male disadvantage in mortality and relevant premature diseases [69]. Similar to our results, sex-specific differences and organ-specific responses were observed in enzymatic and non-enzymatic antioxidant metabolic pathways of ovine twins in the fetal brain, liver and skeletal muscle, with GR activity and GSH lower in the livers of female fetuses than in male fetuses, while no significant changes were observed between SOD, CAT and GSH-Px [71]. Male fetuses are more susceptible to a wide range of complications associated with oxidative stress, while the female profile is associated with a different set of diseases, all of which are likely due to their sex-specific GSH metabolism [69]. Female fetuses may be more susceptible to oxidative stress-induced growth restriction and neurodevelopmental disorders, which can lead to anxiety and affective disorders later in the life cycle [71,72]. Further research on the sex-specific antioxidant defense system and its response to oxidative stress during the prenatal period is needed to better understand the impact on health and development.

In this study, the organ-specific mechanism of Se use was confirmed by analyzing the response of the liver and placenta (previously published) to Se exposure. In our previous study, increased bioavailability and toxicity of SeNPs compared to NaSe were observed in the placenta [26]. Opposite to maternal liver response to SeNPs administration, severe oxidative stress in the placenta negatively affected prenatal development and had long-term consequences. The observed embryo or fetal lethality and reduced body weight of fetuses that completed development represent the result of a direct effect of SeNPs on the fetus, and also the outcome of altered placental function indirectly affecting fetal development [26]. 

Otherwise, mechanisms of controlled placental endocytosis of selenoprotein P or GSH-Px3 pinocytosis are responsible for the transport of a sufficient amount of Se from the maternal to the fetal circulation when maternal Se levels are adequate [13]. After placental passage, blood travels through the umbilical vein and enters the fetal liver, which therefore has a high Se uptake from the mother. Se status in fetal livers was significantly increased in both groups of Se-exposed fetuses (1.6-fold for SeNPs and 1.5-fold for NaSe compared to the controls), but the differences in maternal Se levels in liver and blood, where SeNPs exceeded NaSe-induced accumulation, are no longer present. To protect the fetus from Se overexposure, the placenta accumulated twice as much Se after SeNPs administration compared to NaSe, creating a suboptimal physiological environment [26]. The fetal liver as a key organ for controlling conditions in utero did not change histologically after Se exposure, but the antioxidant defense system responded depending on fetal sex and the applied Se forms.

Differences between males and females in the baseline level of the antioxidant defense system influence the response to Se exposure. Females have the same response pattern to both forms of Se, while males show a spectrum of responses, from no response to SeNPs to an intense response to selenite. The parameters extracted by PCA that separated male NaSe were GSH-Px, GSH and GST (factor 1) and SH, CAT and GR (factor 2), which were the major contributors.

## 5. Conclusions

In conclusion, the present study provides a detailed insight into the differences in maternal and fetal liver antioxidant responses to treatment with Se nanoparticles compared to the most commonly used inorganic sodium selenite, such as differences in bioavailability, differential effects on pregnant females and fetuses, and the sex-specific responses in fetuses. Based on the SeNPs formulation, the redox imbalance in the liver of pregnant females was less pronounced after the use of SeNPs than NaSe, although the accumulation of Se in the liver was twice as high in this group. It has been confirmed that during pregnancy, urinary excretion of Se is the main mechanism to eliminate the excess and reach the optimal concentration necessary for normal physiological functioning. Moreover, SeNPs have an advantage over NaSe depending on the stage of the life cycle: SeNPs have a more pronounced positive effect on oxidative stress parameters in the liver of pregnant females than NaSe. On the other hand, prenatal development was negatively affected by SeNPs in contrast to NaSe. Finally, the inherent sex differences in intact fetuses were complemented by the different response pattern of the liver to both forms of Se: while female fetuses show a similar response pattern to both forms of Se, male fetuses exhibit a spectrum of responses ranging from no response to SeNPs to an intense response to selenite.

## Figures and Tables

**Figure 1 antioxidants-13-00756-f001:**
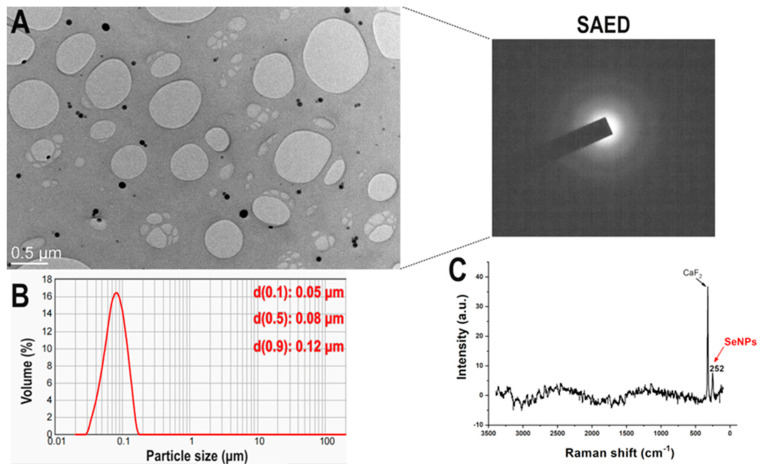
Physicochemical characterization of SeNPs. (**A**)—Representative TEM image with SAED of the SeNPs synthesized by the chemical reduction method; (**B**)—particle size distribution of these SeNPs and their (**C**)—Raman spectrum.

**Figure 2 antioxidants-13-00756-f002:**
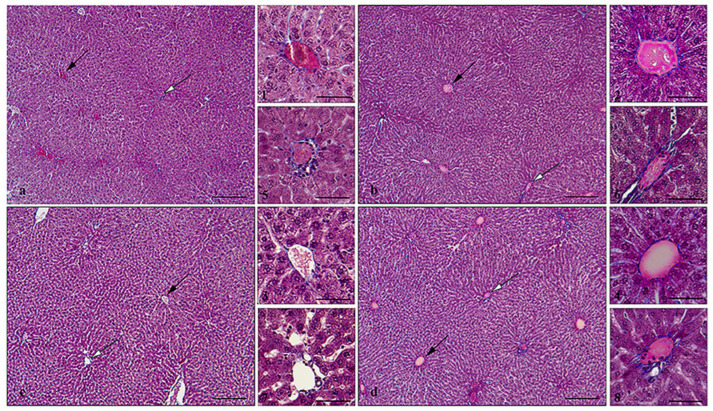
Representative micrographs of livers of intact control (IC) (**a**); control (C) (**b**); SeNPs-treated pregnant female liver (**c**); and NaSe-treated pregnant female liver (**d**). Masson’s trichrome stain; central vein—black arrow and insets (**1**–**4**); portal triad—white arrow and insets (**5**–**8**); bar = 200 µm, bar in insets = 50 µm.

**Figure 3 antioxidants-13-00756-f003:**
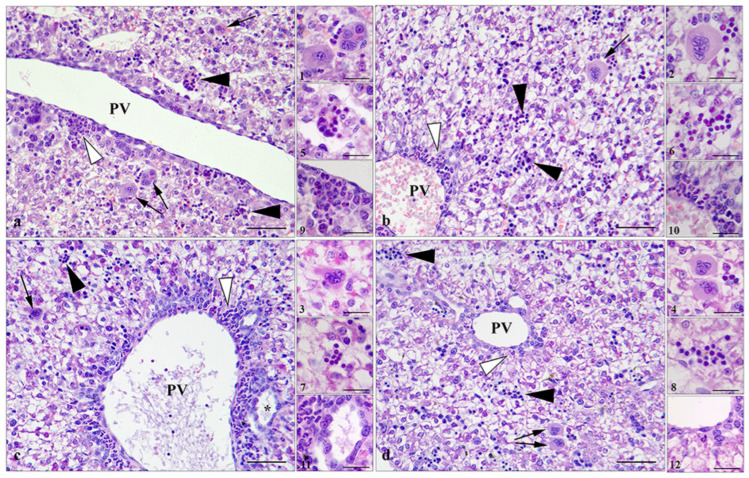
Representative micrographs of the liver of near-term fetuses of control (**a**); control (**b**); SeNPs-treated mothers (**c**); and NaSe-treated mothers (**d**). Portal vein—PV; megakaryocytes—arrow and (**1**–**4**) insets; islands of erythropoiesis—black arrowhead and (**5**–**8**) insets; islands of myelopoiesis—white arrowhead and (**9**–**12**) insets; bile duct—asterisk and inset 11; bar = 50 µm, bar in insets = 20 µm.

**Figure 4 antioxidants-13-00756-f004:**
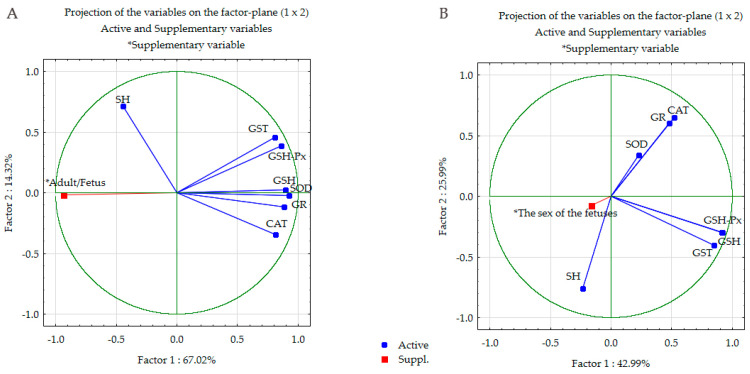
The PCA shows the contribution of investigated oxidative stress parameters in differences between the livers of pregnant females and fetuses of both sexes—(**A**), and between female fetuses and male fetuses—(**B**). n = 6 pregnant females in each group; n = 20 fetuses of both sexes in each group (10 females and 10 males).

**Figure 5 antioxidants-13-00756-f005:**
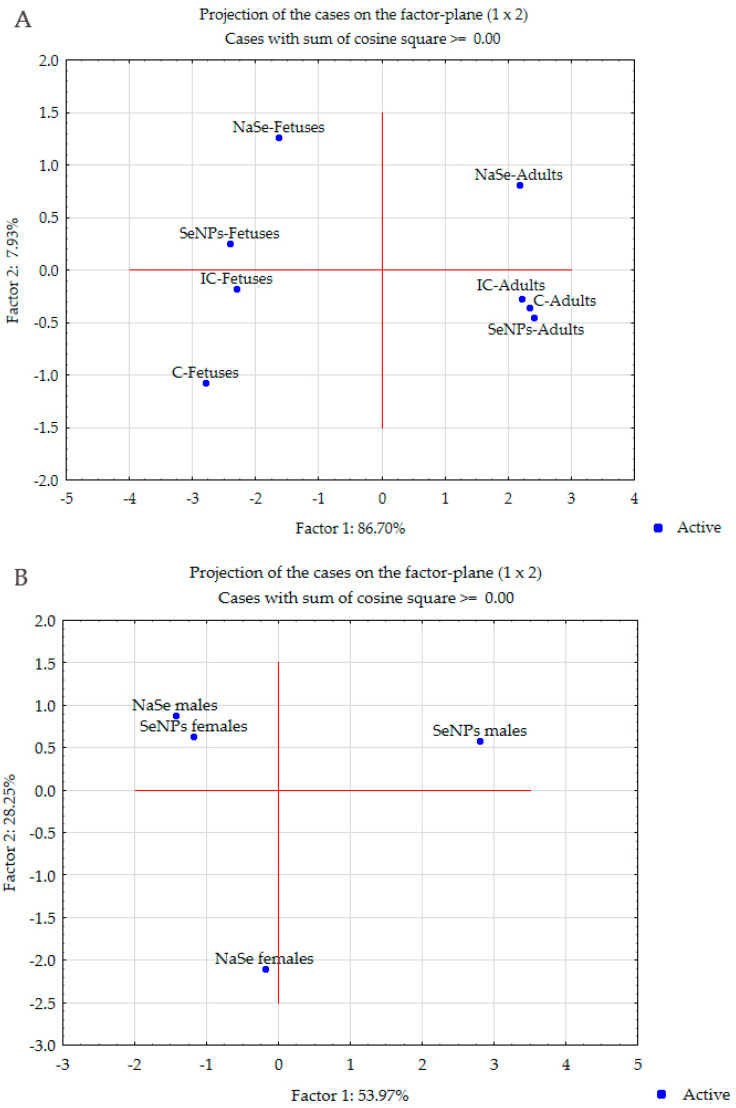
(**A**) Projection of the cases on the factor plane between pregnant females and both female and male fetuses. (**B**) Projection of the cases on the factor plane between female and male fetuses. The label for the group is in front of the parameter name: IC—intact control; C—control; SeNPs—nano-selenium particles; NaSe—sodium selenite. N = 6 pregnant females in each group; N = 20 fetuses of both sexes in each group (10 females and 10 males).

**Table 1 antioxidants-13-00756-t001:** Chemical composition of the rat food used in the experiment.

Chemical Composition of Rat Diet					
General		Vitamins		Minerals	
Proteins	min 20.00%	Vit A	min 12,000 U/kg	Zinc	min 30 mg/kg
Fat	min 5.00%	Vit D3	min 2000 U/kg	Copper	min 15.00 mg/kg
Humidity	max 11.5%	Vit E	min 30.00 mg/kg	Iron	min 60 mg/kg
Cellulose	max 5.00%	Vit B1	min 4.00 mg/kg	Manganese	min 60 mg/kg
Ashes	max 8.00%	Vit B2	min 7.00 mg/kg	Iodine	min 0.15 mg/kg
Calcium	min 0.95%	Vit B6	min 6 mg/kg	Selenium	min 0.30 mg/kg
Phosphorus	min 0.70%	Vit B12	min 0.05 mg/kg		
Lysine	min 0.60%	Folic acid	min 1.0 mg/kg	**Antioxidants**	
Methionine	min 0.60%	Choline chloride	1000.00 mg/kg	E-321, E-320, E-338, E-202	min 125 mg/kg
Tryptophan	max 0.15%	Niacin	20.00 mg/kg		
		Pantothenic acid	min 10.00 mg/kg		

**Table 2 antioxidants-13-00756-t002:** The activities of superoxide dismutase (SOD, U/g wet mass), catalase (CAT, µmol H_2_O_2_/min/g wet mass), glutathione peroxidase (GSH-Px, nmol NADPH/min/g wet mass), glutathione reductase (GR, nmol NADPH/min/g wet mass) and glutathione S-transferase (GST, nmol GSH/min/g wet mass), and concentrations of sulfhydryl groups (SH, µmol/g wet mass) and total glutathione (GSH, nmol/g wet mass) in the liver of pregnant females and fetuses. Differences in the liver between adult females and fetuses of both sexes. Data are given as Mean ± SD. Nonparametric one-way ANOVA and the factorial ANOVA were used. The Tukey HSD test was used to seek the differences between and within groups. *p* < 0.05 was considered significant. N = 6 pregnant females in each group; N = 20 fetuses of both sexes in each group (10 females and 10 males).

Liver	Pregnant Females				Fetuses of Both Sexes			
	IC	C	SeNPs	NaSe	IC	C	SeNPs	NaSe
SOD	9191.8 ± 512.8	9201.6 ± 490.1	8672.1 ± 532.4	7617.9 ± 179.2 ^A,B^	1064.2 ± 88.6 *	896.4 ± 264.2 *	1057.2 ± 195.3 *	1065.3 ± 188.4 *
CAT	58,008.8 ± 10,160.2	66,186.6 ± 5854.9	50,194.7 ± 9565.4	32,832.2 ± 12,866.9 ^A,B,C^	18,154.7 ± 8467.6 *	14,121.9 ± 3028.1 *	11,065.1 ± 2780.7 *	25,649.7 ± 10,748.3 ^A,B,C,^*
GSH-Px	133,021.3 ± 4077.4	112,770 ± 9.0	134,472.3 ± 3489.8	152,585.2 ± 26,756.4 ^A,B,C^	7493.3 ± 1366.2 *	10,658.8 ± 3006.0 *	9177.8 ± 1600.2 *	6395.3 ± 5376.4 *
GR	6764.5 ± 543.6	6947.5 ± 214.4	8170.4 ± 638.9 ^A,B^	8059.0 ± 656.4 ^A,B^	3872.9 ± 1024.9 *	4257.2 ± 878.5 *	4244.4 ± 564.6 *	5175.1 ± 1070.6 ^A,B,C,^*
GST	232,897.8 ± 5529.2	221,184.7 ± 34,092.1	270,279.2 ± 38,519.2 ^B^	259,164.6 ± 5,746,468.3 ^B^	58,372.0 ± 53,309.3 *	50,220.2 ± 9777.9 *	51,604.2 ± 6376.7 *	46,846.6 ± 12,238.7 *
SH	3118.3 ± 97.2	3093.4 ± 358.8	3418.1 ± 362.1	2566.9 ± 368.4 ^C^	4167.5 ± 788.1 *	5135.9 ± 1358.0 ^A,^*	3948.8 ± 395.9 ^B,^*	2957.2 ± 730.6 ^A,B,C,^*
GSH	3574.2 ± 2347.8	4032.9 ± 525.3	3713.3 ± 473.2	2983.9 ± 443.0	878.2 ± 186.4 *	282.4 ± 177.9 ^A^*	316.3 ± 252.9 ^A,^*	320.9 ± 172.6 ^A,^*

^A^ In respect to IC; ^B^ in respect to C; ^C^ in respect to SeNPs; * pregnant females vs. fetuses; abbreviations: IC—intact control; C—control; SeNPs—selenium-nanoparticle-treated animals; NaSe—sodium-selenite-treated animals.

**Table 3 antioxidants-13-00756-t003:** The activities of superoxide dismutase (SOD, U/g wet mass), catalase (CAT, µmol H_2_O_2_/min/g wet mass), glutathione peroxidase (GSH-Px, nmol NADPH/min/g wet mass), glutathione reductase (GR, nmol NADPH/min/g wet mass) and glutathione S-transferase (GST, nmol GSH/min/g wet mass), and concentrations of sulfhydryl groups (SH, µmol/g wet mass) and total glutathione (GSH, nmol/g wet mass) in the liver of female and male fetuses. Data are given as Mean ± SD. Nonparametric one-way ANOVA and the factorial ANOVA were used. The Tukey HSD test was used to seek the differences between and within groups. *p* < 0.05 was considered significant. N = 20 fetuses of both sexes in each group (10 females and 10 males).

Liver	Female Fetuses				Male Fetuses			
	IC	C	SeNPs	NaSe	IC	C	SeNPs	NaSe
SOD	1038.9 ± 108.	782.2 ± 245.7 ^A^	1095.9 ± 170.2 ^B^	1050.6 ± 219.5 ^B^	1083.2 ± 72.3	1101.9 ± 154.8 *	1010.7 ± 232.8	1076.7 ± 169.1
CAT	14,752.5 ± 1345.2	25,662.4 ± 8840.2 ^A^	11,976.1 ± 3155.9 ^B^	14,625.5 ± 3255.3 ^B,^*	20,706.4 ± 10,700.6 *	13,215.7 ± 2647.6 ^A,^*	9971.9 ± 2035.4 ^A^	25,639.8 ± 12,376.8 ^A,B,C,^*
GSH-Px	6883.8 ± 928.	10,578.5 ± 3828.5 ^A^	8054.2 ± 1154.7	12,314.4 ± 333.5 ^A,C,^*	30,075.3 ± 423.2 *	10,803.2 ± 110.9	10,526.0 ± 754.4	18,420.1 ± 1120.6 ^A,B,C,^*
GR	3329.2 ± 414.9	4085.1 ± 959.6	4527.3 ± 276.5	5578.7 ± 1019.5 ^A,B^	4280.8 ± 1176.5 *	4566.7 ± 692.9	3904.8 ± 661.2	4864.7 ± 1039.9
GST	40,467.0 ± 16,662.3	53,459.5 ± 10,968.5	50,913.2 ± 7820.8	43,482.1 ± 6216.9	71,800.8 ± 67,812.8 *	44,389.6 ± 2002.1 *	52,433.3 ± 4860.4	49,434.8 ± 15,130.3 ^A,B,C^
SH	4245.9 ± 759.9	5468.8 ± 1615.53	3929.6 ± 399.2 ^B^	2795.1 ± 506.7 ^A,B^	4108.6 ± 855.4	4536.7 ± 275.3	971.7 ± 437.7 *	3081.9 ± 864.4 ^A,B^
GSH	405.4 ± 130.1	239.3 ± 148.9	448.3 ± 276.2	291.8 ± 179.9	1232.8 ± 1505.5 *	360.0 ± 216.3 ^A^	157.9 ± 84.6 ^A,^*	343.2 ± 170.0 ^A^

^A^ In respect to IC; ^B^ in respect to C; ^C^ in respect to SeNPs. * males vs. female. Abbreviations: IC—intact control; C—control; SeNPs—selenium-nanoparticle-treated animals; NaSe—Sodium-selenite-treated animals.

**Table 4 antioxidants-13-00756-t004:** The concentration of Se in the blood, urine and feces of pregnant females collected from: 1. intact control (IC); 2. control (C); 3. selenium-nanoparticle-treated animals (SeNPs); and 4. sodium-selenite-treated animals (NaSe). Data are given as Mean ± SD. The differences between groups were tested with the nonparametric Mann–Whitney U test. *p* < 0.05 was considered significant. N = 6 pregnant females in each group.

Se	IC	C	SeNPs	NaSe
Blood (µg/L)	177.59 ± 76.00	139.22 ± 19.85	1705.32 ± 737.33 ^A,B^	283.79 ± 153.69 ^A,B,C^
Urin (µg/L)	30.45 ± 23.05	279.25 ± 267.81 ^A^	1802.11 ± 1093.23 ^A,B^	433.57 ± 129.91 ^A,B,C^
Feces (µg/kg)	195.53 ± 182.45	241.68 ± 222.25	196.03 ± 116.40	167.88 ± 112.88

^A^ In respect to IC; ^B^ in respect to C; ^C^ in respect to SeNPs.

**Table 5 antioxidants-13-00756-t005:** The concentration of selenium in the livers of pregnant females and fetuses of both sexes of: 1. intact control (IC); 2. control (C); 3. selenium-nanoparticle-treated animals (SeNPs), and 4. Sodium-selenite-treated animals (NaSe). Data are given as Mean ± SD. The differences between groups were tested with the nonparametric Mann–Whitney U test. *p* < 0.05 was considered significant. N = 6 pregnant females in each group; N = 20 fetuses of both sexes in each group (10 females and 10 males).

Liver Se (µg/kg)	IC	C	SeNPs	NaSe
Pregnant females	895.95 ± 233.32	1404.51 ± 339.42 ^A^	2846.33 ± 382.3 ^A,B^	1543.81 ± 316.69 ^A,C^
Fetuses	136.25 ± 12.97 *	252.75 ± 57.01 ^A,^*	410.89 ± 150.90 ^A,B,^*	382.66 ± 194.75 ^A,B,^*

^A^ In respect to IC; ^B^ in respect to C; ^C^ in respect to SeNPs. * Pregnant females vs. fetuses.

**Table 6 antioxidants-13-00756-t006:** The concentration of Se in the livers of male and female fetuses of: 1. intact control (IC); 2. control (C); 3. selenium-nanoparticle-treated animals (SeNPs); and 4. sodium-selenite-treated animals (NaSe). Data are given as Mean ± SD. The differences between groups were tested with the nonparametric Mannn–Whitney U test. *p* < 0.05 was considered significant. N = 20 fetuses of both sexes in each group (10 females and 10 males).

Se (µg/kg)	IC	C	SeNPs	NaSe
Male	125.45 ± 0.15	208.91 ± 0.10 ^A^	434.54 ± 197.67 ^A,B^	333.49 ± 99.32 ^A,B,C^
Female	147.05 ± 8.38	267.36 ± 59.94 ^A,^*	387.24 ± 127.18 ^A,B^	431.83 ± 268.68 ^A,B,^*

^A^ In respect to IC; ^B^ in respect to C; ^C^ in respect to SeNPs. * Males vs. females.

**Table 7 antioxidants-13-00756-t007:** Contribution of variables to the changes between treatments in pregnant females, fetuses of both sexes, female fetuses and male fetuses.

	Adults vs. Fetuses of Both Sexes		Female Fetuses vs. Male Fetuses	
Variables	Factor 1	Factor 2	Factor 1	Factor 2
SOD	0.181810 *	0.000537	0.017298	0.062483
CAT	0.140249	0.119641	0.091023	0.233260 *
GSH-Px	0.158244	0.149126 *	0.279709 *	0.049327
GR	0.167017 *	0.013317	0.076629	0.201821 *
GST	0.139691	0.206296 *	0.239921 *	0.088416
SH	0.042649	0.510407 *	0.018033	0.316676 *
GSH	0.170340 *	0.000677	0.277387 *	0.048017

* parameters that contributed the most.

## Data Availability

The data presented in this study are available on a reasonable request from the corresponding author.

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
