# Peer review of "Antioxidant Response of Maternal and Fetal Rat Liver to Selenium Nanoparticle Supplementation Compared to Sodium Selenite: Sex Differences between Fetuses"

_antioxidants, 2024, doi:10.3390/antiox13070756_

Round 1
Reviewer 1 Report
Figure 2: Images are blurry and the quality is very low. Moreover, higher magnifications (can be showed as insert) are needed
Figure 3: Higher magnifications are necessary (those indicated by the arrows for example)
In this manuscript authors compared the effects of organic selenium nanoparticles and inorganic sodium selenite on the antioxidant response in maternal and fetal rat liver. Authors found that the structure of the liver of gravid females remains histologically the same after supplementation with both forms of Se, while the oxidative stress in the liver was significantly lower after the use of SeNPs compared to NaSe.
the manuscript is interesting and generally well written. Tables are clear but figures need improvement (see comments below). The topic is interesting and fits the aim of the journal. However, some important improvements are necessary. My comments are listed below.
Introduction: since oxidative stress plays a critical role in this manuscript, its effects deserve to be highlighted. In fact, oxidative stress plays a key role in the occurrence and progression of several diseases including cnacer (PMID: 36641100, PMID: 35901941) and pregnancy outcome (see PMID: 37296665 ). This is an important point to add since it can highlight the important results obtained by the authors.
Figure 2: Images are blurry and the quality is very low. Moreover, higher magnifications (can be showed as insert) are needed
Figure 3: Higher magnifications are necessary (those indicated by the arrows for example)
Table 2: Abbreviations should be moved at the end of the table
Figure legends: Add the number of replicates (N) in the legend of each figure
Fix superscripts and subscripts throughout the manuscript
Author Response
Response to Reviewer #1:
Major Comments:
Reviewer’s comment 1: Introduction: since oxidative stress plays a critical role in this manuscript, its effects deserve to be highlighted. Oxidative stress plays a key role in the occurrence and progression of several diseases including cancer (PMID: 36641100, PMID: 35901941) and pregnancy outcomes (see PMID: 37296665). This is an important point to add since it can highlight the important results obtained by the authors.
Reply to comment 1: As suggested by Reviewer #1, we have included in the text of the Introduction section the basics about oxidative stress and its effects on the development and progression of some diseases such as cancer and the effects on pregnancy outcomes caused by preeclampsia. In doing so, we have cited the suggested articles as relevant literature sources. Consequently, we added 5 new references (now [16-20] in the References section, Lines 892-901).
Lines 83-102: we have inserted the following text:
“Many studies have focused on investigating the link between oxidative stress as a trigger of genetic and epigenetic dysregulation of oncogenes that can promote neoplastic transformation and tumorigenesis. An important modulator in the response to oxidative stress is nuclear factor erythroid 2-related factor 2 (NRF2), which induces the expression of antioxidant enzymes and thus protects cells from oxidative damage. The NRF2/KEAP1 signaling pathway in cervical and endometrial cancer and chemotherapy resistance can be modulated by various natural and synthetic antioxidant modulators (Tossetta and Marzioni, 2023) [16]. The same authors (Tossetta and Marzioni, 2022) [17] suggested a wide range of natural modulators of NRF2, such as diosmetin, wedelolactone, epoximicin, ailantone, Vit K3, caffeic acid, and many others. Therefore, treatments with natural or synthetic compounds that can inhibit NRF2 signaling could be an important clinical step to combat or suppress drug resistance in ovarian cancer patients (Tossetta and Marzioni, 2022) [17]. Selenium exhibits a broad spectrum of contradictory properties in living organisms. The chemopreventive, cancer-promoting and anticancer effects of Se are three different aspects of the complicated relationship of Se to cancer (Ratan et al., 2022) [18], which likely depends on its chemical form and bioavailability. In addition, preeclampsia, which can occur during pregnancy and has a direct effect on the outcome of the pregnancy is also characterized by increased oxidative stress and inflammation, in which NRF2 also plays a key role (Tosseta et al., 2023) [19]. The study by Vanderlelie and Perkins (2011) [20] has shown that Se supplementation can help reduce oxidative stress in women at risk of preeclampsia.“
Reviewer’s comment 2: Figure 2: Images are blurry and the quality is very low. Moreover, higher magnifications (which can be shown as an insert) are needed.
Reply to comment 2: We have significantly improved the quality of Figure 2, removed the blurring, and presented a higher magnification as an insert.
Reviewer’s comment 3: Figure 3: Higher magnifications are necessary (those indicated by the arrows for example).
Reply to comment 3: We have presented higher magnification in Figure 3 at the places marked with an arrow.
Reviewer’s comment 4: Table 2: Abbreviations should be moved to the end of the table.
Reply to comment 4: As recommended by Reviewer #1, abbreviations are moved at the end of Table 2, as well as Table 3.
Reviewer’s comment 5: Figure legends: Add the number of replicates (N) in the legend of each figure.
Reply to comment 5: We have added the number of replicates (N) in the legend of each table and figure.
Reviewer’s comment 6: Fix superscripts and subscripts throughout the manuscript.
Reply to comment 6: We have carefully checked the entire text and fixed the superscript and subscript letters (Lines 232, 257, 709, 710 and 712).

Reviewer 2 Report
In this study, the authors investigate the antioxidant response of dams and their fetuses to selenium nanoparticle supplementation compared to sodium selenite. For the study, selenium nanoparticles and sodium selenite were either synthesized or acquired, and administered to an equivalent amount to the pregnant dams. Experimentally, female Wistar rats were primarily used for impregnation. Animals were maintained in proper controlled conditions, and were divided into 4 groups: 1 = Intact Control; 2 = Control plus BSA and vit. C; 3 = group with SeNPs at the dose of 0.5 mg SeNP/kg/body weight; 4 = group with NaSe administered at the concentration as SeNPs. All treatment were administered from da 1 of gestation until day 21 of gestation, when the pregnant dams and the fetuses were sacrificed, tissues were collected and tissue histology as well as several antioxidant biochemical parameters (SOD, CAT, GSH-Px, GR, GST, total GSH, and SH) were assessed. For the fetuses, differences between male and female fetuses were also recorded for further analysis. Data were analyzed by ANOVA followed by Tukey's test.
The results reported in the study indicate that no differences were observed between treatment with SeNPs and NaSe. Marked differences were observed between the animals in group 3 as compared to animals in group 4. Animals in these two groups presented with major differences as compared to animals in groups 1 and 2. In particular, the activity of SOD was reduced in the liver of the NaSe groups compared to animals in groups 1 and 2. The activity of CAT was also significant decreased along the same lines. . The activity of GSH-Px was increased in animals in group 3 as compared to groups 1 and 2, and 3. The activity of GR, on the other hand, was significantly increased in animals in group 3 and 4 as compared to groups 1 and 2. The activity of GSH was increased in group 3 and 4 as compared to group 2 only SH levels were decreased in animals in group 4 compared to animals in group 3 No changes in GSH levels between adult animals across the gorups were noted. When assessing similar parameters in the fetuses, SOD, GSH-Px and GSH levels presented no differences regardless of the group considered. CAT and GR activities were significantly increased in the fetuses form group 4 ad comparedto the other three groups. The concentration of SH groups was increased in group 2 vs. group 1, but was lower in fetuses from group 3 as compared to group 2. The levels of SH in fetuses form group 4 were lower when compared to the other three groups. GSH concentration was lower in animals in group 2, 3, and 4 as compared to 1. Differences in oxidative tress parameters were further differentiated by the authors based on the gender of the fetuses in a given group.
Specific Comments:
The manuscript is properly written and can be properly followed. The study is properly conducted and the conclusions provided by the authors are in line with the data reported in the manuscript.
Experimentally: The number of adult animals (i.e. pregnant dams) used per group is not properly indicated in the text or in the tables and figures. At line 125, the text reports a ratio of 2 female in estrus per male, but it is unclear if this the whole number of female per group of the grouping per cage within a certain experimental group. Similarly, it is not reported how many pups per groups were analyzed and what was the gender distribution within a group. Because of the use of ANOVA, it is suspected that the number of animals per group was sizeable but having the numbers reported in the manuscript will add emphasis and significance of the data reported.
Experimentally: It doesn't appear that the authors investigated the antioxidant parameters in the male parent. Assessment of these parameters in the males adult Wistar rats would have provided an important comparison vs. the male fetuses, especially in lieu of differences between male and female fetuses.
Results: in Table 2 and 3 the units of the various enzymatic parameters are not reported.
Discussion: The discussion is appropriate and the relevance of one type of Se supplementation vs. the other is properly discussed.
Typos: A few typos were noticed. For example: line 280: at least 1 word is missing between ...used and vibrational...
Author Response
Response to Reviewer #2:
Experimentally:
Reviewer’s comment 1: The number of adult animals (i.e. pregnant dams) used per group is not properly indicated in the text or the tables and figures. In line 125, the text reports a ratio of 2 females in estrus per male, but it is unclear if this is the whole number of females per group of the grouping per cage within a certain experimental group. Similarly, it is not reported how many pups per group were analyzed and what was the gender distribution within a group. Because of the use of ANOVA, it is suspected that the number of animals per group was sizeable but having the numbers reported in the manuscript will add emphasis and significance of the data reported.
Reply to comment 1: We thank the Reviewer for this important comment. We have added the number of animals used per group, both pregnant females and fetuses, in the Materials and methods section and the Tables and Figure legends. The use of two females and one fertile male is a description of the mating procedure, after which the sperm-positive females were randomly assigned to one of four experimental groups so that there were 6 pregnant females in each group (IC, C, SeNPs and NaSe) (Lines 146-147). We also calculated the average number of fetuses and the sex distribution (Lines 156-158). The average litter size was 12±2 pups i.e. fetuses, with equal gender distribution, average number of males as well as females was 6±1. The four experimental groups formed with male (IC, C, N, S) and female (IC, C, N, S) fetuses each comprising 10 fetuses, whereby care was taken to ensure that the fetuses came from different mothers. To make the text clearer, it was reformulated (Lines 164-168).
Reviewer’s comment 2: It doesn't appear that the authors investigated the antioxidant parameters in the male parent. Assessment of these parameters in the male adult Wistar rats would have provided an important comparison vs. the male fetuses, especially in lieu of differences between male and female fetuses.
Reply to comment 2: We thank the Reviewer for his valuable commentary. As the Reviewer indicated, selenium plays a very important role in male reproduction, which should be related to the condition of the fetus, and serious research should be devoted to this topic. However, the main objective of our study was to analyze and compare the effects of selenium supplementation during pregnancy with two forms of selenium (nanoselenium and selenite) on mothers and fetuses with possible sex-specific effects on fetal livers. The study was therefore designed as a developmental toxicity study and not as a reproductive toxicity study in which we would analyze the effects of selenium treatment on male reproductive health. Additionally, data on the effects of nanoparticles (of different types) on prenatal development are very scarce, and the existing results show that they are very specific. In addition, observing the relationship between mother and fetus provides valuable insights into the complex effects of nanoparticles, which, as we have seen, are highly dependent on the physiological environment (pregnancy and development).
Results:
Reviewer’s comment 3: In Tables 2 and 3 the units of the various enzymatic parameters are not reported.
Reply to comment 3: In Materials and Methods, the units of the parameters studied are defined, but as suggested by Reviewer #2, we have also included the units in Tables 2 and 3.
Typos:
Reviewer’s comment 4: A few typos were noticed. For example: in line 280: at least 1 word is missing between ...used and vibrational...
Reply to comment 4: Many thanks to Reviewer #2 for carefully reading the text. We have reworded the entire sentence in which the Reviewer discovered the error. We have also carefully checked the entire manuscript for typographical and printing errors.
The new sentence (Lines 307-309):
„Although Raman spectroscopy is a less common vibrational spectroscopy technique, it is very useful for the qualitative analysis of inorganic materials, which usually exhibit weak bands in the lower wavenumber range“.

Round 2
Reviewer 1 Report
the manuscript has been significantly improved after revision and can be accepted in the current form
none